# Strong Polyamide-6 Nanocomposites with Cellulose Nanofibers Mediated by Green Solvent Mixtures

**DOI:** 10.3390/nano11082127

**Published:** 2021-08-20

**Authors:** Pruthvi K. Sridhara, Ferran Masso, Peter Olsén, Fabiola Vilaseca

**Affiliations:** 1Advanced Biomaterials and Nanotechnology, Department of Chemical Engineering, University of Girona, 17003 Girona, Spain; pruthvi.sridhara@udg.edu; 2Department of Fiber and Polymer Technology, KTH Royal Institute of Technology, 10044 Stockholm, Sweden; ferran.masso@gmail.com (F.M.); polsen@kth.se (P.O.)

**Keywords:** cellulose nanofiber, polyamide-6, solvent casting, mechanical properties

## Abstract

Cellulose nanofiber (CNF) as a bio-based reinforcement has attracted tremendous interests in engineering polymer composites. This study developed a sustainable approach to reinforce polyamide-6 or nylon-6 (PA6) with CNFs through solvent casting in formic acid/water mixtures. The methodology provides an energy-efficient pathway towards well-dispersed high-CNF content PA6 biocomposites. Nanocomposite formulations up to 50 wt.% of CNFs were prepared, and excellent improvements in the tensile properties were observed, with an increase in the elastic modulus from 1.5 to 4.2 GPa, and in the tensile strength from 46.3 to 124 MPa. The experimental tensile values were compared with the analytical values obtained by micromechanical models. Fractured surfaces were observed using scanning electron microscopy to examine the interface morphology. FTIR revealed strong hydrogen bonding at the interface, and the thermal parameters were determined using TGA and DSC, where the nanocomposites’ crystallinity tended to reduce with the increase in the CNF content. In addition, nanocomposites showed good thermomechanical stability for all formulations. Overall, this work provides a facile fabrication pathway for high-CNF content nanocomposites of PA6 for high-performance and advanced material applications.

## 1. Introduction

Plant-based reinforcement is attractive for enhancing the mechanical properties of polymers in the context of biodegradable and sustainable materials [1]. The reinforcing effect of plant-based fibers originates from the highly crystalline cellulose hierarchical structure. Cellulose is both of a renewable origin and biodegradable [2]. In the nanocellulose form, cellulose exhibits an extraordinary potential as a reinforcing element in composites due to its high aspect ratio and high specific strength combined with its low density. It is also possible to chemically modify its surface to tailor the properties in applications such as foams, filter media films, adhesives, hierarchical materials, and electronic materials [3]. When well dispersed, nanocellulose exhibits a higher specific surface area, facilitating interface adhesion with the polymeric matrix which enables efficient stress transfer. Among the plant-based nanocelluloses, there are two major categories: cellulose nanocrystals (CNCs) and cellulose nanofibers (CNFs). Typically, CNCs are highly crystalline, needle-like structures, with a few hundred nanometers in length and a few nanometers in width [4]. On the other hand, CNFs are fibril-like structures that contain both crystalline and amorphous phases, with diameters in the order of tens of nanometers and lengths typically ranging from tens to hundreds of micrometers [5]. The linearity of cellulose polymer chains and their strong intermolecular bonds enables the formation of ordered crystalline structures, which impart exceptional mechanical properties to CNFs [6]. The extraction process of CNFs from natural fibers is an essential factor as the properties of CNFs depend mainly on the source of material and the method of extraction. CNFs are produced via mechanical defibrillation, often facilitated by either chemical or enzymatic pretreatments to reduce the energy consumption of the process [7]. The inherent properties of CNFs make them an interesting sustainable choice of reinforcement for polymeric matrices intended for the automotive, construction, packaging, and energy sectors [8].

Polyamides as thermoplastic matrix materials with CNF reinforcement provide superior mechanical and thermal properties of composites. The most common type of polyamide is polyamide-6 (PA6, also called nylon-6). This polymorphic, biodegradable, and bio-compatible thermoplastic polymer is an attractive choice for its availability, higher mechanical properties, and compatibility with CNFs. Due to the hydrophilic nature of both PA6 and CNFs, good interface interactions are formed without the use of a compatibilizer or coupling reagents [9]. The compatibility of PA6 and CNFs results in an excellent dispersion, distribution, and interfacial adhesion. However, silane coupling agents can improve the adhesion between a polyamide matrix and CNFs [10].

Recent studies have shown increments in mechanical properties for composites of PA6 with cellulose pulp fibers and/or micro-fibrillated cellulose (MFC) through melt processing with the help of a high-speed thermo-kinetic mixer [11,12]. MFCs are more heterogeneous and larger in size than CNFs. As CNFs offer a larger specific surface area for the polymer to form strong and functional interface adhesion, the overall reinforcing potential of CNFs is greater than that of MFCs [13]. One of the drawbacks of melt processing PA6/cellulose composites is the high melting point of PA6 (~220 °C), which leads to the onset of thermal degradation of cellulose during processing. Techniques such as solvent casting circumvent thermal degradation, retaining the thermal stability of cellulose, and facilitate uniform CNF dispersion in the matrix [14]. The primary advantage of solvent casting is the ease of fabrication without specialized equipment that is essential for other techniques such as extrusion and injection molding processes. The cast film exhibits a homogenous thickness distribution and fiber dispersion [15]. Currently, the most commonly used solvent is dimethylformamide (DMF). DMF is aprotic by nature, has a high dielectric constant and low volatility, and provides a good dispersion of CNFs in PA6 matrices. However, DMF is a potent liver toxin and may cause abdominal problems and reduce sperm motility [16]. A green solvent alternative would be a significant leap forward. In this study, we used formic acid which is renewable and has a lower toxicity than DMF. Previous studies have reported formic acid as an excellent solvent for dissolving PA6. Formic acid dissolves PA6 at 30 °C with negligible degradation of the polymer, promoted by the weak acid characteristics [17,18]. CNFs have good dispersion in water [19]. Therefore, we considered a water/formic acid mixed system to both dissolve PA6 and disperse the CNFs. This study developed a green solvent casting system based on water and formic acid to produce high-CNF content PA6 nanocomposites. The nanocomposites were produced by diluting the CNF dispersion in water in formic acid containing 20 wt.% of dissolved PA6 corresponding to each formulation. The homogenous mixture was then dried to produce thin films for characterization.

## 2. Materials and Methods

### 2.1. Materials

An industrial grade of PA6 called Ultramid^®^ B3S (density *ρ* = 1.13 g/m^3^) was commercially purchased from BASF SE (Ludwigshafen, Germany) in the form of pellets. This grade of PA6 is colorless, less viscous, and suitable for producing thin-walled technical parts. The pellets were powdered from Powder Plastics Europe SL (Valls, Spain) and then passed through a 1000-micron sieve to reduce the dissolving time of PA6 in formic acid (Sigma Aldrich 33015-1L-M, St. Louis, MO, USA). The grain size distribution of PA6 is shown in Table 1. Commercially available cellulose pulp provided by Nordic Paper Säffle AB (Säffle, Sweden) was used to derive the CNFs. This product was of a high cellulose content, with 87% of pure cellulose, 12 to 12.5% of hemicellulose, and less than 1% of lignin. CNFs were then used to reinforce the PA6 matrix.

### 2.2. Methods

#### 2.2.1. Preparation of PA6-CNF Nanocomposite Films

The cellulose pulp was disintegrated into cellulose nanofibers (CNFs) by an enzymatic pretreatment described by Henriksson et al., 2007, using an endoglucanase enzyme, namely, Novozyme 476 (Novozymes AS, Bagsværd, Denmark) [20]. Enzymatic treatment provided a more specific, milder, and environmentally friendly method to increase the yield and expedite the disintegration of cellulose [21,22]. Firstly, the cellulose pulp fibers were beaten mechanically in a PFI mill (Hamjern, Hamar, Norway) at 1000 revolutions to increase the swelling in water and provide better accessibility of the enzyme to the cellulose. Further, enzymatic treatment was carried out by dispersing 3 wt.% pulp in 50 mM tris(hydroxymethyl)aminomethane/HCl buffer with pH 7 and 0.02 wt.% enzyme relative to the amount of pulp. The fibers were incubated for 2 h at 50 °C and later washed on a Büchner funnel. The fibers were kept for 30 min at 80 °C to cease the enzyme activity and later washed again. Then, the fibers were beaten in the PFI mill at 4000 revolutions. Following the enzyme pretreatment, 1.5 wt.% fiber suspension in water was prepared and subjected to homogenization (Laboratory Homogenizer 15M, Gaulin Corp., Boston, MA, USA) at a constant high pressure without heating. Initially, the suspension was at room temperature, but with an increasing number of passes, the temperature increased slightly, facilitating the homogenization of the CNFs [23]. The fibers were passed through the homogenizer six times until the suspension reached a consistent viscosity based on visual inspection to obtain a CNF gel with 1.5 wt.%. consistency. Previous work in our group by Prakobna et al. determined that these CNFs have an average diameter ranging from 6 to 9 nm and lengths in the 0.7–2 µm range [24].

The Petri dish used for casting the PA6-CNF nanocomposites had an inner diameter of 8.8 cm. To produce thin films of thickness 0.1 mm, an approximate total weight of 0.8 g of the PA6-CNF mixture was required. The amount of PA6 powder required for each composition of cellulose was calculated using the density, mass, and volume relation. For obtaining 0.8 g of nanocomposite film, the weight specifications of CNF and PA6 are shown in Table 2.

For every formulation, 20 wt.% of PA6 was dissolved completely in formic acid at 30 °C. The 1.5 wt.% CNF gel was then dispersed within formic acid to obtain a consistency of 0.75 wt.% CNF content in the solvent system. This pre-wetting of CNFs in the same solvent was conducted in order to further improve CNF/PA6 compatibility and their interfacial adhesion [15]. To ensure proper dispersion of CNFs, the solution was mixed with the help of an Ultraturrax (T 25, IKA, Königswinter, Germany) by stirring at a high speed of 7000 rpm for about 30 s. This solution was then left in an Ultrasonic bath (Exibel, Clas Ohlson, Dalarna, Sweden) for 10 min to remove any possible bubbles. Finally, the CNF gel dispersed in formic acid and the dissolved PA6 in formic acid were mixed with the help of a stirrer for 2 h to ensure uniform mixing. This mixture was again left in the ultrasonic bath to remove any bubbles which might have been formed during mixing. The mixed solution was introduced onto the Petri dish carefully and placed in an oven (Memmert UM200, Büchenbach, Germany) at 50 °C for casting the PA6-CNF nanocomposite. When all the formic acid and water had evaporated, we placed the Petri dish containing the composite in a desiccator at ambient temperature to prevent deformation while cooling. The composite film was removed from the Petri dish with the help of a spatula and tweezers. Rectangular samples were cut using a hydraulic press (Stans & Press—TJT Teknik AB, Vilshult, Sweden) from the thin film for tensile and DMA characterizations. An example of the cast nanocomposite film (50 wt.%) is shown in Figure 1. The neat CNF film was produced by the filtration method. PA6 films were also prepared to be used as control samples.

#### 2.2.2. Tensile Tests

Rectangular samples of dimensions (*l* × *b*) 45 mm × 6 mm were cut from the films to perform tensile tests. The thickness values of each nanocomposite sample were measured using a digital precision micrometer (Starrett, Athol, MA, USA). Prior to tensile testing, the rectangular samples were dried overnight in an oven at 50 °C and then conditioned at room temperature of 23 ± 2 °C and 50 ± 5% relative humidity (RH) for 48 h. The uniaxial tensile properties were measured on a Universal Testing Machine (Instron 5944, Norwood, MA, USA) where the extension was quantified with a high-precision camera. Tensile tests were performed with a 500 N load cell and strain rate of 0.1 min^−1^. The measured tensile strength and modulus were averaged over at least five samples for statistical significance. For comparative assessment, the tensile strength of the nanocomposites was also estimated by a basic rule of mixtures (ROM) model:(1)$$\sigma_{NC}=\sigma_{CNF}\;V_{CNF}+\sigma_{PA6}\;\left(1-V_{CNF}\right)$$
where *σ_NC_* is the tensile strength of the nanocomposite, *σ_CNF_* and *σ_PA_*_6_ are the experimental tensile strengths of CNFs and PA6, respectively, and *V_CNF_* is the total volume fraction of CNFs. The value of *σ_NC_* predicts the linear relationship between *σ_NC_* and the CNF composition. An estimation of reinforcement efficiency for all the formulations of CNF content was resolved. All volume fractions were calculated assuming the densities of 1.13 g/m^3^ for PA6 and 1.5 g/m^3^ for CNFs.

The Cox–Krenchel micromechanical model was used to predict the tensile modulus. This model was developed based on the classical shear lag theory and is one of the most widely used models [1]. Assumptions used in this model are: (i) the fiber and matrix respond elastically, (ii) no axial loads on the fiber ends, and (iii) an ideal fiber matrix interface. The Cox–Krenchel model is defined as
(2)$$E_{NC}=\eta_{0}V_{CNF}E_{CNF}\left(1-\frac{tanh\left(ns\right)}{ns}\right)+\left(1-V_{CNF}\right)E_{PA6}$$
where *n* is expressed by
(3)$$n=\sqrt{2E_{PA6}/\left[E_{CNF}\left(1+V_{PA6}\right)ln\left(\frac{1}{V_{CNF}}\right)\right]}$$

In the above Equations (2) and (3), *E_NC_* is the elastic modulus of the nanocomposite, *E_CNF_* and *E_PA_*_6_ are the experimental moduli of CNFs and PA6, respectively, with the fiber orientation factor *η*_0_ = 3/8 assuming an in-plane isotropic orientation of fibers in a random short fiber polymer composite, and *s* is the fiber aspect ratio where the weight average fiber length *L* can be used to calculate *s* = *L*/*D*, where *D* is the diameter of the fiber [25].

Similarly, elastic moduli were calculated assuming a random in-plane fiber orientation using the Tsai–Pagano model defined in Equation (4) [26].
(4)$$E_{NC}=\left(\frac{3}{8}\right)\;E_{L}+\left(\frac{5}{8}\right)\;E_{T}$$
(5)$$E_{L}=E_{CNF}\;V_{CNF}+E_{PA6}\;\left(1-V_{CNF}\right)$$
(6)$$E_{T}=\frac{E_{CNF}\;E_{PA6}}{E_{CNF}\;\left(1-V_{CNF}\right)+E_{PA6}\;V_{CNF}}$$
where *E_L_* and *E_T_* are the longitudinal and the transverse modulus of the nanocomposite calculated longitudinally and transversely to the direction of the fiber assuming a unidirectional composite with cylindrical fibers. *E_NC_* is the theoretical nanocomposite modulus for a random in-plane fiber orientation.

#### 2.2.3. Scanning Electron Microscopy (SEM)

The morphology of samples fractured due to elongation during the tensile tests was observed under a scanning electron microscope (Hitachi S4800, Ibaraki, Japan) to characterize the interaction between CNFs and the PA6 matrix. The samples were observed at an accelerated voltage of 15.0 kV and a short working distance. The cross-sections of the fractured samples were coated with a thin layer (2–4 nm) of Pt:Pd with the help of a Cressington sputter prior to observation.

#### 2.2.4. Fourier Transform Infrared Spectroscopy (FTIR)

The changes in the chemical structure and the binding configuration of the nanocomposite samples were analyzed by using Fourier transform infrared analysis. The FTIR spectrum of the films produced was obtained in transmission by performing the analysis in the IR frequency range of 500 to 4000 cm^−1^ using a Perkin Elmer Spectrum 100 spectrometer (PerkinElmer, Waltham, MA, USA).

#### 2.2.5. Thermogravimetric Analysis (TGA)

The control samples and all the formulations of PA6-CNF nanocomposites were analyzed for thermal degradation using a TGA1 STAR^e^ System (Mettler Toledo, Greifensee, Switzerland) at the heating rate of 10 °C/min. The tests were carried out by placing the sample in an open platinum pan within a nitrogen environment and heating it from 30 to 600 °C.

#### 2.2.6. Differential Scanning Calorimetry (DSC)

The thermal properties of the PA6 (powder and control film) and PA6-CNF nanocomposite films were measured using a DSC 1 STAR^e^ System (Mettler Toledo, Greifensee, Switzerland). The DSC was run three times for each sample. The samples were firstly heated from 30 to 260 °C and held at this temperature for 2 min followed by cooling of samples to 30 °C within a nitrogen environment. The heating and cooling rate was 10 °C/min. The degree of crystallinity of the polymer after processing was determined from the enthalpy corresponding to the melting endotherm of the first heating, according to Equations (7) and (8).
(7)$$\Delta H_{polymer}=\Delta H_{sample}\cdot \left(\frac{1}{1-\frac{w_{CNF}}{100}}\right)$$
(8)$$\chi_{c}=\frac{\Delta H_{polymer}}{\Delta H_{100\%}}\cdot 100$$
where $\Delta H_{sample}$ is the enthalpy of the melting endotherm of the sample from the first heating cycle, and $\Delta H_{polymer}$ is the melting enthalpy of the PA6 polymer in the sample. The degree of crystallinity ($\chi_{c}$) considers the melting enthalpy for the 100% crystalline PA6, $\Delta H_{100\%}$ = 230 J/g [27].

#### 2.2.7. Dynamic Mechanical Analysis (DMA)

Rectangular samples of dimensions (*l* × *b*) 15 mm × 6 mm were cut from neat PA6 and PA6-CNF nanocomposite casted films to perform DMA in tensile mode. These samples were kept in the oven for 48 h at 55 °C prior to analysis. The analysis was performed using the instrument DMA Q800 (TA Instruments, New Castle, DE, USA) measuring from 30 °C and finishing at 160 °C, with a heating rate of 5 °C/min, preload of 1 N, a constant frequency of 1 Hz, and strain amplitude of 0.1%. The DMA tests were run three times for each formulation.

## 3. Results and Discussions

### 3.1. Morphology

The PA6 and CNF nanocomposite films produced via solvent casting had a good dispersion and distribution for all formulations, which was evident from the uniform translucency observed throughout the films. There were no visual signs of voids or agglomeration even for the high-CNF composition, thus confirming a good phase morphology. The films were generally flat and were easy to cut into rectangular samples for testing. The pure CNF film was obtained through the filtration process and had only small pores. As the sample thickness significantly affects the calculated mechanical properties, the thickness measurement was conducted with a digital precision micrometer. The mean thicknesses of all rectangular samples for each formulation used for tensile testing are shown in Table 3.

### 3.2. Tensile Tests

The tensile strengths and elastic moduli for all the formulations are summarized in Table 3. During tensile testing, there were no failures at the grips due to the clamping pressure. The tensile properties of the nanocomposites increased with the CNF content, corroborating that the CNFs were well dispersed within the PA6 matrix. Previous studies have demonstrated that carboxyl groups increase during enzymatic pretreatment [28]. These carboxyl groups play a vital role in the dispersion of CNFs within a matrix. The repulsion due to negative charges on the surface of the CNFs prevents fibril–fibril agglomeration, aiding in achieving a uniform distribution and consequently contributing to the mechanical properties of the nanocomposite films. The CNFs imparted intrinsic mechanical properties to the nanocomposites. Both the tensile strength and Young’s modulus are appreciably higher, indicating good wetting and effective stress transfer at the PA6–CNF interface [29]. The uniform dispersion and distribution of CNFs provide a high surface area that increases the number of secondary interactions with the PA6 matrix [30]. The highest tensile stress at break and Young’s modulus were observed for the 50 wt.% formulation nanocomposite with the values of 124 MPa and 4.2 GPa, respectively, corresponding to an improvement of 2.7 times when compared to the PA6 control sample. For the high-CNF content nanocomposites, casting/evaporation causes strong interactions between the CNFs, promoting strong percolating network formation through hydrogen bonding. The casting process ensures that the PA6 matrix penetrates this CNF network formation and enhances the mechanical properties. For such high-CNF content films, the improvement in tensile properties is dependent on the CNF network followed by successful penetration of the PA6 matrix and may not be attributed to the intrinsic stiffness of individual nanoparticles. For the neat CNF film, the maximum tensile strength of ~201 MPa and modulus of 12.23 GPa were obtained.

Previously, we prepared up to 25 wt.% CNF formulations with PA6 by melt compounding using a Gelimat thermo-kinetic mixer and observed good enhancement in mechanical properties [31]. Peng et al. obtained composite formulations up to 10 wt.% CNF content with PA6 via thermal compounding using a Brabender with minor improvements in the tensile properties [32]. Lee et al. prepared 40 wt.% CNF/PA6 by silane treatment and the calendaring process, achieving ~2.5 GPa and ~12.5 MPa for the tensile modulus and strength, respectively [33]. Joshi et al. obtained up to 50 wt.% of regenerated cellulose composite membranes with PA6 using the electrospinning method with tensile strength and modulus increments corresponding to ~12% and ~150%, respectively [34]. More relatively, Qua et al. used the solution casting technique to prepare PA6 and 5 wt.% flax and micro-crystalline cellulose (MCC) nanofibers obtained from acid hydrolysis, and the composite’s tensile strength was improved drastically, which is promising for our study [35]. Our study develops an energy-efficient approach towards solvent casting by adopting simple green solvent mixtures to fabricate PA6 nanocomposites using CNFs derived enzymatically, which is a more environmentally friendly way to extract CNFs when compared to other processes such as acid hydrolysis and TEMPO oxidation. In addition, tensile mechanical improvements of about 176% (for modulus) and 168% (for strength), when compared to neat PA6, were achieved for the 50 wt.% formulation without using any dispersion or coupling agents.

The strain at break decreased with an increase in the CNF composition. The CNFs provide larger surface areas for the PA6 matrix to form strong interfacial adhesion, enabling the effective stress transfer and imparting stiffness to the composite [36]. Our previous work with PA6 and cellulosic material biocomposites produced via melt processing showed a considerable drop in strain levels for 25 wt.% cellulose pulp [12,31]. Similarly, in this study, strain at break decreased with the increase in the CNF content. This behavior is inherent to a well-dispersed CNF network where the CNF behavior is retained in the corresponding CNF-polymer matrix nanocomposites [37]. CNFs hinder the nanocomposites’ elongation due to stiffening caused by the reinforcing effect. Nevertheless, despite the higher CNF contents, an elongation of ~7% was measured for the 50 wt.% formulation. Figure 2 shows the stress–strain plots and the deformation behavior during elongation for all the samples.

As an approximate assessment of nanocomposites’ tensile strength *(σ_NC_*), we used the basic rule of mixtures model (ROM). ROM assumes that *σ_NC_* depends on the strength of the CNF nanopaper (*σ_CNF_*) and neat PA6 (*σ_PA_*_6_), scaled with the corresponding fiber volume fractions. The prediction from Equation (1) shows the nanocomposites’ lower bound tensile strength as the tensile strength of a single CNF is assumed to be underutilized. Experimental values ranging from 103 to 232 MPa for the cellulose nanopaper were recorded in the previous literature [38,39]. The difference in tensile strength is due to the nanofiber and nanofiber networks [40]. Based on the ROM model, the experimental values of the tensile strength match reasonably well with the predicted values for all the CNF composition formulations (Figure 3A). Overall, for high-fiber volume fraction nanocomposites, the predicted values should be considered purely theoretical since the nanocellulose might not exist as a dense nanopaper network but as a 3D percolated network of CNFs [1].

Figure 3B shows the experimental elastic moduli for nanocomposites with moduli calculated from the Cox–Krenchel equations, Equations (2) and (3), and the Tsai–Pagano equations, Equations (4)–(6). The Tsai–Pagano model overestimates the elastic moduli of the nanocomposites except for two formulations (30 wt.% and 40 wt.%), suggesting that the stiffness of the cellulose nanofiber network was controlling the moduli of the nanocomposites. The Cox–Krenchel model considers the aspect ratio of CNFs and is a good method to approximate the effect on the nanocomposites’ elastic modulus. As the aspect ratio of nanofibers decreases, the fiber ends act more effectively as stress and strain fields in the PA6 matrix due to the discontinuity [41]. The experimental values are similar to the estimated values for low-CNF composition nanocomposites (5 wt.% and 10 wt.%), whereas they are higher than the theoretical values with CNF compositions of 20 wt.%, 30 wt.%, and 50 wt.%. This suggests there was good impregnation of the PA6 matrix within the CNF network.

### 3.3. SEM Analysis

The morphology of the fractured tensile cross-sections of the 20 wt.% and 50 wt.% CNF nanocomposites and neat CNF nanopaper was analyzed with a scanning electron microscope. Figure 4 shows the SEM micrographs of the fractured surfaces at three different magnification levels. The distribution of CNFs appears to be homogenous within the in-plane layers at the nanoscale, with very few voids and no agglomerations for the nanocomposite films (Figure 4A,D). Typically, nanocomposites’ fractured surfaces had a brittle fracture with deformation occurring in different planes. Under tensile load, the different moduli of PA6 and CNFs lead to different modes of deformation, generating different strains which create stress concentration points at the CNF–PA6 interface. When this unmatched strain surpasses the magnitude of interfacial adhesion, a sliding deformation occurs [31]. The resulting surface morphology due to this sliding deformation is seen in Figure 4B,E. As the CNF content increased, the surface morphology of the nanocomposites transitioned to be partially identical to that of the pure CNF nanopaper (Figure 4G). The fractured surfaces for the nanocomposites were formed due to elongation; hence, the PA6 strands have a fiber-like shape which have been fractured at some point under tension. These PA6 strands are adjoined by CNFs, indicating good adhesion, and the strands should not be confused for agglomerated fibers (Figure 4C,E). The dispersion of fibers can be observed from Figure 4B,C,E,F with strands of PA6 surrounded by tiny fibrils of CNFs. For the CNF nanopaper, the hydroxyl groups on the CNFs result in hydrogen bonding between individual CNFs, hence creating a 3D percolated network, as seen in Figure 4H,I.

### 3.4. FTIR Analysis

FTIR was used to confirm the hydrogen bonding behavior and crystallinity in the neat samples and PA6-CNF nanocomposites [42]. The FTIR spectra for the neat samples and 40 wt.% and 50 wt.% nanocomposite formulations are shown in Figure 5. The pure PA6 film had absorptions at 3323 cm^−1^ (hydrogen-bonded N-H stretching vibration), 3038 cm^−1^ (N-H in-plane bending), 2930 cm^−1^ (stretching vibration CH2), 1634 cm^−1^ (stretching vibration C=O), 1537 cm^−1^ (stretching vibration C-N and CO-N-H bend), 1463 cm^−1^ (CH2 in-plane bending), 966 cm^−1^ (stretching vibration C-CO), and 693 cm^−1^ (out-of-plane bending N-H) [43]. The PA6 polymer can effortlessly undergo a phase transition from the γ to the α form because of external inducement or modifications in the processing parameters as there are minor differences in energy between the α and γ forms. Hence, the structure and crystalline morphology are important to the PA6 properties [44]. The IR spectra below 1500 cm^−1^ are sensitive to the polymorph structure and are used as a spectroscopic marker to identify the presence of contrasting crystalline forms in the sample [45]. The sharp increase in marker bands (Figure 5B) correlates with the α form at 693 cm^−1^, 920 cm^−1^, 1200 cm^−1^, and 1418 cm^−1^ in relation to the marker bands at 1172 cm^−1^. The crystalline α-phase PA6 has a planar zig-zag structure and is thermodynamically more stable, while the γ phase has a helix structure and is metastable [34]. Since the CH_2_ group is not ideally packed, hydrogen bonds are more efficient in the α form. Contrastingly, in the γ form, hydrogen bonding is dominated by the ideal packing of methylene blocks [46]. Interactions lead to the formation of new hydrogen bonds with the amide groups of PA6 and the OH groups of cellulose. The intensity of the marker peaks after 1030 cm^−1^, at lower-CNF composition nanocomposites, signifies the C-O stretching. The verge at 1722 cm^−1^ correlates with a C=O stretching vibration resulting from a small degree of esterification of the cellulose OH groups [47]. The affinity of CNFs and PA6 reorients the original hydrogen bond in the PA6 polymer chain. The reorientation of the hydrogen bonds is more evident for nanocomposites with a higher CNF content.

### 3.5. TGA Analysis

Figure 6 shows the thermogravimetric (TGA) and the first derivative thermogravimetric (DTG) curves for the control samples and PA6-CNF nanocomposites. For the control samples, single-stage thermal decomposition bands with a distinct maximum weight loss peak were observed. However, for the nanocomposite films, the thermal bands seemed to have two-stage decomposition with two peaks indicating maximum weight loss. The two-stage thermal decomposition is clearly observed in Figure 6B for the 30 wt.% and 40 wt.% formulations. Degradation initiation of PA6 occurs at ~405 °C, while for the nanocomposites, it starts at ~290 °C (Figure 6A). During thermal degradation of CNFs above 290 °C, cellulose decomposes, releasing combustible volatiles such as acetaldehyde, propenal, methanol, and acetic acid. These volatile products accelerate the decomposition rate in the samples [34,48]. The degradation bands for the PA6-CNF nanocomposites were between the control samples’ degradation bands (Figure 6B). From the DTG curves, maximum degradation temperatures of ~440 °C and ~350 °C were observed for the PA6 and CNF films, respectively. Overall, the presence of the CNF content in the nanocomposites reduces the thermal stability of the nanocomposites.

### 3.6. DSC Analysis

The polymer’s melting and crystallization behavior in the PA6 powder, PA6 film, and PA6-CNF nanocomposite films was analyzed with DSC. The glass transition temperature (Tg), melting temperature (Tm), crystallization temperature (Tc), and the degree of crystallinity (*χ_c_*) of the PA6 polymer for all formulations are summarized in Table 4.

The DSC thermograms corresponding to the first heating and cooling cycles appear in Figure 7. There was no morphological change in the crystal structure of the polymer between the PA6 powder and the PA6 film (Figure 7A) since the Tm was similar in both cases (~223 °C), which corresponded to the α-form and γ-form PA6 crystals [49]. A tendency for decreased Tm with the increase in the CNF content was observed because of a reduction in the crystalline order. The crystallinity of the PA6 pellets was slightly greater than that of the PA6 casted film. This change in crystallinity suggests that formic acid evaporation results in the formation of a metastable crystalline structure [50]. With the addition of CNFs, the Tg for the nanocomposites occurred at a moderately higher temperature compared to the control PA6 film. This increase in Tg is a result of the good dispersion of CNFs within the PA6 matrix and the strong interfacial adhesion [51,52]. As the CNF content increased, the crystallinity of nanocomposites reduced. Strong interactions between the polar amide groups of PA6 and the hydroxyl groups of CNFs resulted in constrained segmental mobility of the PA6 chain [53]. This restricted conformational freedom for the PA6 polymer and reduced the crystallinity of the nanocomposites.

On cooling, a distinct crystallization exothermic peak (Figure 7B) for neat PA6 at ~194 °C relates to the α-form crystal [54]. Tc’s tendency to decrease as the CNF content increased means that crystallization of PA6 begins later in the nanocomposites when compared to neat PA6. Cellulosic materials at lower concentrations usually serve as nucleating agents that increase the crystallization rate in PA6 [55,56]. However, at high weight fractions of CNFs, the crystallization rate is retarded, reducing the degree of crystallinity, as observed in our nanocomposites. Overall, in our case, it can be stated that the mobility of PA6 chains is restricted by the addition of CNFs, thus reducing the crystallinity.

### 3.7. DMA Tests

Dynamic mechanical analysis was used for characterizing the viscoelastic behavior of neat PA6 and PA6-CNF nanocomposites by a sinusoidal deformation in tensile mode. Due to the polymer’s viscoelastic nature, the stress and strain for such materials will represent a damping factor (*δ*). The reliance of the storage modulus (*E*′) and tan *δ* at temperatures between 30 and 150 °C for neat PA6 and the nanocomposite formulations is shown in Figure 8. From Figure 8A, the nanocomposite samples show good thermomechanical stability with an increase in the CNF composition. The improved thermomechanical properties of the nanocomposites indicate the presence of good interfacial bonding between the CNFs and the PA6 matrix [57]. With the increase in temperature, PA6 showed a drastic decrease in the storage modulus. The CNFs had a remarkable effect on the nanocomposites’ viscoelastic behavior, with an increase in the storage modulus throughout the range of temperatures with the addition of CNFs. The presence of nanofibers reduces the elongation and promotes stiffness, aiding the stress transfer through the nanocomposites [58]. The increase in the storage modulus for the higher-CNF content nanocomposites suggests the presence of percolated network structures of cellulose in the PA6 matrix, constraining the long-range motion of polymer chains and resulting in a pseudo-solid behavior [30]. The presence of nanofibers reduces the elongation and promotes stiffness, aiding the stress transfer through the nanocomposites [58]. Since the storage modulus serves as an accurate measure of molecular rigidity, it shows that the nanocomposites become more rigid with the addition of CNFs [59].

Additionally, the tan *δ* intensity for the nanocomposites after the glassy state decreased, as seen in Figure 8B. Traversing from low to high temperatures, the tan *δ* peak for neat PA6 is about ~94 °C (Figure 8B). The values of tan *δ* are related to the glass transition temperature (Tg) and are higher than the values obtained from the DSC analysis. This shift in the Tg is attributed to DMA being more sensitive than DSC as larger specimens are used [60]. Nevertheless, the Tg from the tan *δ* peaks and the DSC results depict similar trends of an increase in the Tg with the addition of CNFs. The reinforcement factor (*F_R_*), defined as the ratio of storage moduli of nanocomposites and control samples, is shown in Table 5 along with the DMA values obtained at the specific temperatures. Expectedly, *F_R_* increases as the CNF composition increases, indicating improved stiffness and corroborating the reinforcement effect.

## 4. Conclusions

Nanocomposite films of PA6, reinforced with a high content of CNFs, were successfully prepared with the solvent casting method using a green solvent mixture of formic acid and water. The CNFs obtained from enzymatic hydrolysis, followed by homogenization, endowed an excellent reinforcement effect, providing a good dispersion, and enabled an excellent stress transfer through the strong interface adhesion between the PA6 and CNFs. The tensile test results demonstrate the excellent reinforcement effect, where the tensile properties for the nanocomposites increased with an increase in the CNF composition. The maximum values of the tensile strength (~124 MPa) and modulus (~4.2 GPa) were obtained with the 50 wt.% CNF formulation. The nanocomposite films were uniform, indicating a good dispersion of CNFs in the PA6 matrix, and no agglomeration was seen in the SEM micrographs. FTIR analysis revealed the molecular interactions between the CNFs and PA6 through hydrogen bonds with the OH groups of cellulose. The TGA results illustrate that with the increase in the CNF composition, the nanocomposites’ thermal stability slightly decreased. DSC revealed small increases in the glass transition, followed by a gradual decrease in crystallinity with the increase in the CNF content as a result of the constrained mobility of PA6 chains upon the addition of CNFs. Maximum crystallinity (~49%) was obtained for the 10 wt.% formulation. The nanocomposites had superior thermomechanical stability with the increase in the CNF composition. This study presents an environmentally friendly method in the interest of high-performance and high-cellulose content bio-nanocomposites.

## Figures and Tables

**Figure 1 nanomaterials-11-02127-f001:**
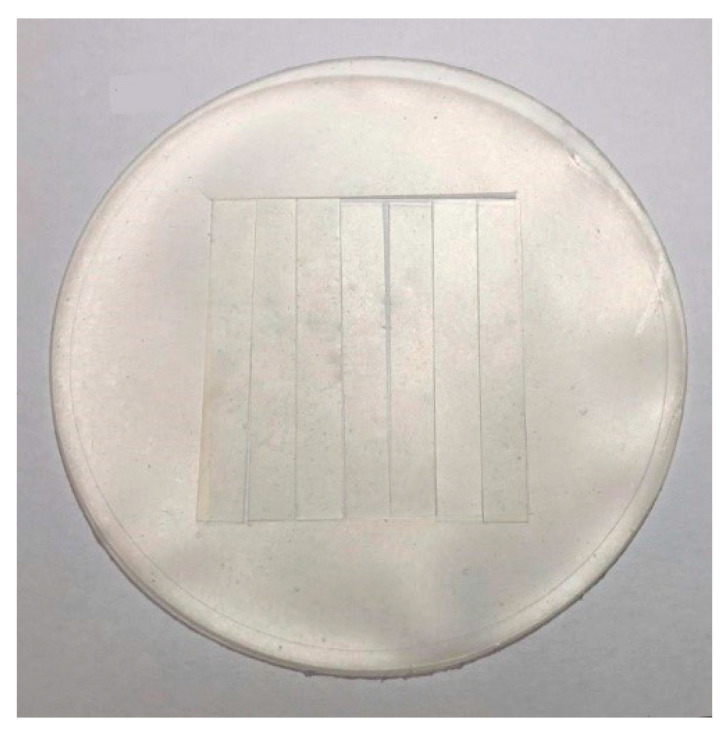
50 wt.% PA6-CNF casted nanocomposite film.

**Figure 2 nanomaterials-11-02127-f002:**
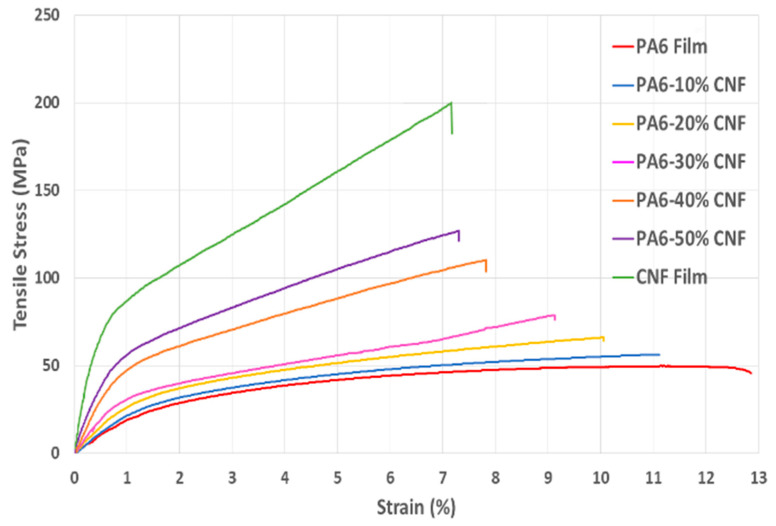
Stress vs. strain curves for control samples and PA6-CNF nanocomposites.

**Figure 3 nanomaterials-11-02127-f003:**
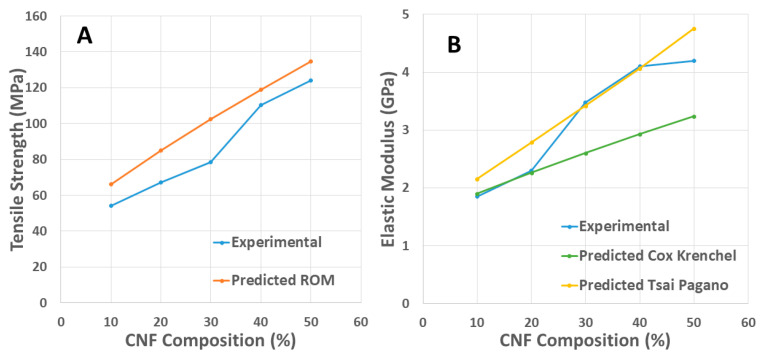
Experimental and theoretically predicted tensile strength (**A**) and modulus (**B**) values for PA6-CNF nanocomposites.

**Figure 4 nanomaterials-11-02127-f004:**
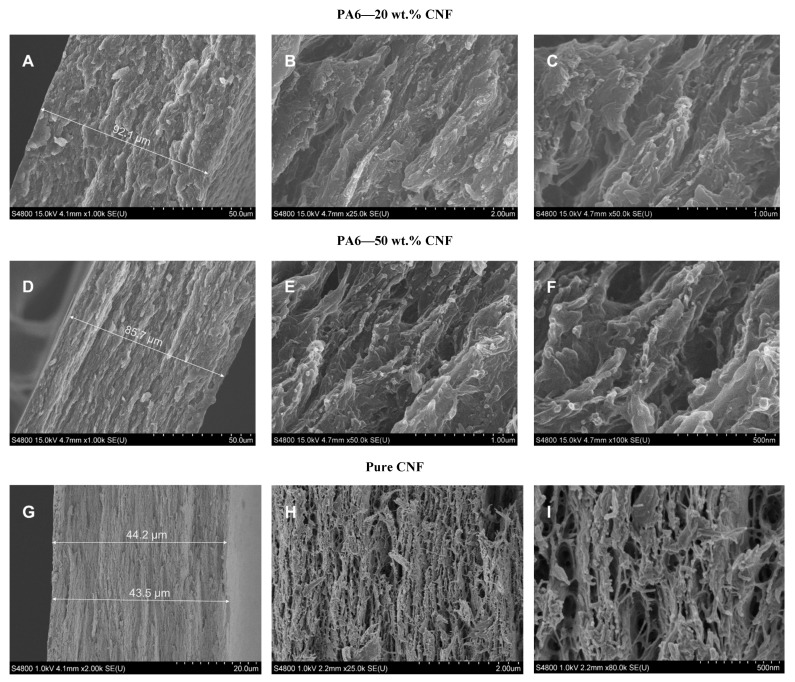
SEM micrographs for 20 wt.% **(A**–**C**), 50 wt.% (**D**–**F**) PA6-CNF nanocomposites, and CNF nanopaper (**G**–**I**).

**Figure 5 nanomaterials-11-02127-f005:**
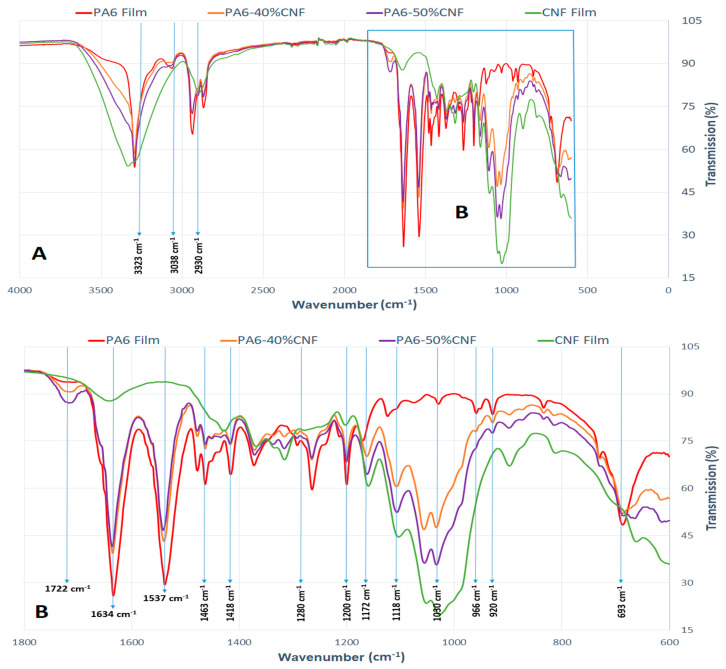
(**A**) FTIR spectra and (**B**) enlarged portion of the FTIR spectra at a lower wavelength for PA6 film, CNF film, and 40% and 50% formulation nanocomposite films.

**Figure 6 nanomaterials-11-02127-f006:**
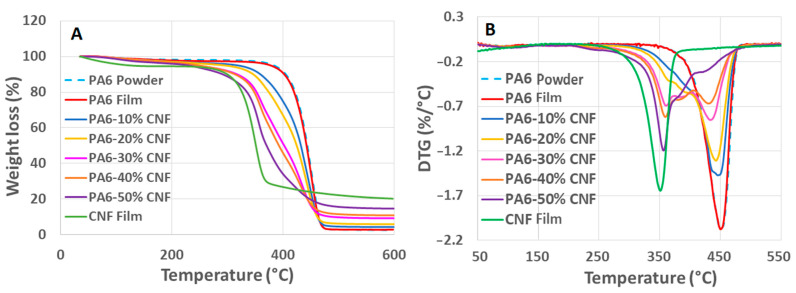
(**A**) TGA curves and (**B**) DTG curves for PA6, CNF, and PA6-CNF nanocomposite films.

**Figure 7 nanomaterials-11-02127-f007:**
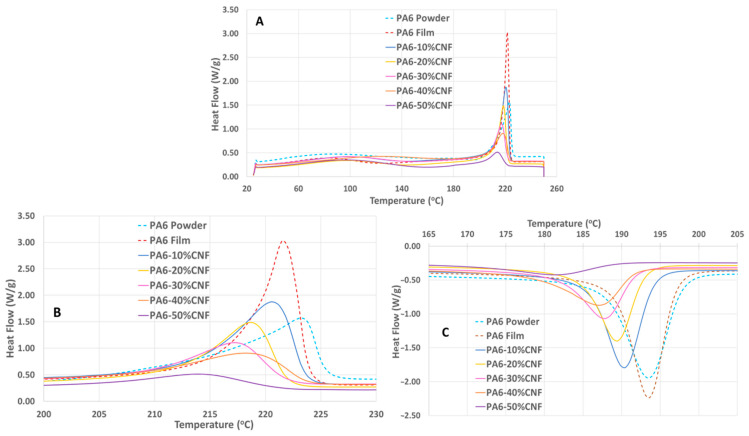
DSC thermograms: (**A**) heating cycle, (**B**) melting region, and (**C**) cooling region, all at a rate of 10 °C/min, for PA6 powder, PA6 film, and PA6-CNF nanocomposite films.

**Figure 8 nanomaterials-11-02127-f008:**
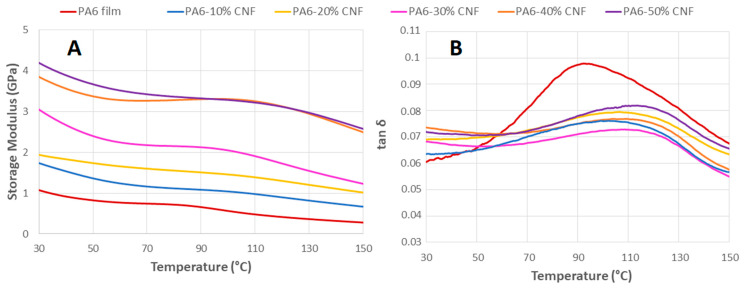
DMA results: (**A**) storage modulus (*E*′), and (**B**) tan *δ* for neat PA6 and the PA6-CNF nanocomposite films.

**Table 1 nanomaterials-11-02127-t001:** Grain size specification and distribution for PA6 powder.

Particle Size Specifications (Micron)	Composition (%)
>1000	0
>800	8.82
>500	33.86
>300	29.42
>100	19.79

**Table 2 nanomaterials-11-02127-t002:** Weight specifications for casting PA6-CNF nanocomposites.

CNF Composition (wt.%)	CNF Dry Weight (g)	CNF Density (g/cm^3^)	PA6 Dry Weight (g)	Composite Density (g/cm^3^)
10	0.08	1.52 ± 0.03	0.72	1.159 ± 0.02
20	0.16		0.64	1.189 ± 0.02
30	0.24		0.56	1.220 ± 0.02
40	0.32		0.48	1.254 ± 0.03
50	0.4		0.4	1.289 ± 0.03

**Table 3 nanomaterials-11-02127-t003:** Tensile properties for the control samples and PA6-CNF nanocomposites.

CNF Composition (wt.%)	Mean Thickness (µm)	Tensile Strength (MPa)	Elastic Modulus (GPa)	Strain at Break (%)
Neat PA6	96.2 ± 8.4	46.3 ± 2.35	1.52 ± 0.08	12.6 ± 0.42
10	111 ± 7.3	54.1 ± 2.87	1.85 ± 0.05	11.1 ± 1.72
20	108.6 ± 1.9	67.3 ± 1.60	2.30 ± 0.08	10.1 ± 1.69
30	116.8 ± 6.3	78.4 ± 0.35	3.48 ± 0.47	9.2 ± 0.49
40	91.3 ± 3.0	110.5 ± 7.80	4.10 ± 0.26	7.8 ± 1.17
50	99.8 ± 2.9	123.9 ± 6.12	4.20 ± 0.27	7.3 ± 0.50
CNF nanopaper	45.4 ± 0.8	200.9 ± 6.73	12.23 ± 0.39	6.9 ± 0.82

**Table 4 nanomaterials-11-02127-t004:** Thermal parameters and degree of crystallinity for the PA6 polymer in the plain matrix and in the PA6-CNF nanocomposites.

CNF Composition (wt.%)	Tg (°C)	Tm (°C)	Tc (°C)	ΔH_Sample_ (J/s)	$\chi_{c}\;(\%)$
PA6 powder	43.4	223.2	193.7	120.4	52.3
0	44.2	223.2	194.0	114.9	49.9
10	48.4	220.3	190.3	101.6	49.1
20	50.2	218.7	189.3	81.4	44.2
30	51.0	217.2	188.0	68.3	42.4
40	53.6	217.0	187.2	54.9	39.8
50	50.9	213.8	182.5	43.7	38.0

**Table 5 nanomaterials-11-02127-t005:** Summary of storage modulus, damping factor, and reinforcement factor (*F_R_*) for neat PA6 and the nanocomposite films at specific temperatures.

CNF Composition (wt.%)	*E*′ at 50 °C (GPa)	*E*′ at 80 °C (GPa)	*E*′ at 120 °C (GPa)	tan *δ* Peaks (°C)	*F_R_* at 50 °C
0	0.82	0.71	0.44	94	-
10	1.35	1.12	0.89	102	1.6
20	1.72	1.55	1.29	108	2.1
30	2.38	2.15	1.71	110	2.9
40	3.36	3.27	3.09	112	4.1
50	3.68	3.36	3.10	117	4.5

## Data Availability

The data presented in this study are available on request from the corresponding author.

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
