# Peer review of "Strong Polyamide-6 Nanocomposites with Cellulose Nanofibers Mediated by Green Solvent Mixtures"

_nanomaterials, 2021, doi:10.3390/nano11082127_

Round 1

Reviewer 1 Report

The authors present a detailed analysis of mechanical properties of cellulose nanofiber-reinforced polyamide composites using a simple but environmentally friendly route for composite preparation. The manuscript is well structured and written. A comparison with the mechanical properties of alike CNF-PA composites prepared by the conventional procedure would be beneficial but is supported by the selected references.

There is just one correction needed. The FTIR spectra in Figure 5 show wavenumbers (cm-1) instead of wavelength (i.e., recirpocal wavenumbers).

Author Response

Reviewer #1

The authors present a detailed analysis of mechanical properties of cellulose nanofiber-reinforced polyamide composites using a simple but environmentally friendly route for composite preparation. The manuscript is well structured and written. A comparison with the mechanical properties of alike CNF-PA composites prepared by the conventional procedure would be beneficial but is supported by the selected references.

            The authors thank the reviewer for the positive remarks. Regarding the comparison of properties of other CNF-PA composites, the authors are planning to prepare a detailed comparison of various CNF-PA composites which are prepared by different methods, as part of another review manuscript.

 There is just one correction needed. The FTIR spectra in Figure 5 show wavenumbers (cm-1) instead of wavelength (i.e., reciprocal wavenumbers).

            The reviewer is true, this is a mistake. The axis title has been corrected to wavenumber (cm-1) in the Figure 5.

Reviewer 2 Report

The paper entitled "Strong Polyamide-6 Nanocomposites with Cellulose Nanofibers Mediated by Green Solvent Mixtures" described the performance  of PA6/Cellulose nanofiber composites. The idea is interesting. The whole manuscript is generally well written. I recommend it to publish on this journal after addressing the following comments.

  1. Line 129, it is better to add the density of CNF in Table 1. In addition, error bar should be added in the composite density column.
  2. Line 420, could you explain why the heating curve of PA6-40wt%CNF had a higher melting peak than PA6-30 wt%CNF? It is quite different form the tredency.
  3. Line 445, the authors claimed that the addition of CNF could improve the storage modulus of the composites. But why the sample PA6-50 wt%CNF did not show a enhancement in storage modulus?
  4. From Figure 8B, we cannot confirm the change tendency of the peak of tan Delta. It is better to add the DMA tests data of PA6-20 wt%CNF and PA6-40 wt%CNF for comparison so that we can have a better discussion about the effects of CNF content on the peak of tan Delta.
  5. Please explain why there are two peaks in the tan Delta curve of PA6-10 wt%CNF.
  6. The data in Table 5 did not match the curves in Figure 8A. Please double check the data.

Author Response

Reviewer #2

The paper entitled "Strong Polyamide-6 Nanocomposites with Cellulose Nanofibers Mediated by Green Solvent Mixtures" described the performance of PA6/Cellulose nanofiber composites. The idea is interesting. The whole manuscript is generally well written. I recommend it to publish on this journal after addressing the following comments.

            The authors appreciate the useful comments of the reviewer in order to improve the manuscript further. All the responses to the comments are mentioned below.

Line 129, it is better to add the density of CNF in Table 1. In addition, error bar should be added in the composite density column.

            The density of CNF and the standard deviation for the composite density are added to the Table 2 (Table 1 refers to size distribution of PA6 powder).

Line 420, could you explain why the heating curve of PA6-40wt%CNF had a higher melting peak than PA6-30 wt%CNF? It is quite different form the tendency.

The authors agree in that "the heating curve of PA6-40wt%CNF had a higher melting peak than PA6-30 wt%CNF". This difference is between 217 degrees (for the 40wt% composition) and 219 degrees (for the 30wt% composition). The authors do not understand why the tendency is slightly different in this case; however, it can be due to local differences in the tested sample.

Line 445, the authors claimed that the addition of CNF could improve the storage modulus of the composites. But why the sample PA6-50 wt%CNF did not show a enhancement in storage modulus?

The Figure 8A has been modified and all the nanocomposites samples are included to show the improvement storage modulus with respect to CNF composition. The authors agree that the storage modulus for the 50 wt.% formulation was not dramatically improved at the higher temperatures. However, at the initial temperature, for the 50 wt.% formulation, the stiffness was kept at higher value than that of 40 wt.% formulation.

From Figure 8B, we cannot confirm the change tendency of the peak of tan Delta. It is better to add the DMA tests data of PA6-20 wt%CNF and PA6-40 wt%CNF for comparison so that we can have a better discussion about the effects of CNF content on the peak of tan Delta.

            The Figure 8B has been modified and all the nanocomposites samples are included to show the tendency of increasing peaks of tan delta with respect to CNF content.

Please explain why there are two peaks in the tan Delta curve of PA6-10 wt%CNF.

            Some polymers can show multiple tan delta peaks and the interpretation varies between systems. Our PA 6 being a semi-crystalline thermoplastic polymer may show different peaks due to the relaxation’s phenomena. In case of polymer composites, the position of the relaxation peaks with respect to the glass transition temperature of the individual components relates with the relative interaction of the components.

            After re-doing the test for all compositions, all the current plots show single tan delta peaks.

The data in Table 5 did not match the curves in Figure 8A. Please double check the data.

            Previously in the Figure 8A, the storage modulus was expressed in logarithmic scale. Now the modified Figure 8A is in actual GPa scale and the Table 5 is revised to match the respective values from the Figure 8.

Round 2

Reviewer 2 Report

I recomment it to be published on this journal.